# In vivo investigation of STN1 downregulation in melanoma formation in adult mice following UV irradiation

Sara Knowles[1], Fang Wang[1], Maarten C. Bosland[2], Shobhan Gaddameedhi[3], Weihang Chai[1,4]*

1 Center for Genetic Diseases, Rosalind Franklin University of Medicine and Science, North Chicago, Illinois, United States of America, 2 Department of Pathology, University of Illinois Chicago College of Medicine, Chicago, Illinois, United States of America, 3 Department of Biological Sciences, North Carolina State University, Raleigh, North Carolina, United States of America, 4 Department of Cancer Biology, Loyola University Chicago Stritch School of Medicine, Maywood, Illinois, United States of America

* weihang.chai@rosalindfranklin.edu

## Abstract

Genome instability is a major force driving tumorigenesis. The ssDNA-binding protein complex CTC1-STN1-TEN1 (CST) plays a pivotal role in maintaining genome stability by countering replication stress, modulating DNA damage repair, and maintaining telomere integrity. Despite its well-documented role in genome maintenance, the involvement of CST in skin cancer development has yet to be investigated. We recently found that CST localizes at stalled DNA replication sites after UV exposure and may suppress the unwanted repriming activity, suggesting a potential role of CST in suppressing genome instability caused by UV damage. In this study, we first analyzed CST expression and alterations in cutaneous melanoma database and found that the CST genes are frequently altered in cutaneous melanoma and their expression is significantly downregulated in melanoma samples compared to normal tissues. We then generated a conditional knockout (cKO) mouse model with STN1 deficiency specifically in melanocytes to investigate its role in skin cancer formation. Upon chronic exposure to UV irradiation, STN1-deficient mice exhibit no obvious difference in melanoma incidence compared to control littermates, suggesting that STN1 downregulation in mature melanocytes has no significant effect on UV-induced skin cancer development in lab mice.

## Introduction

Melanoma is the most aggressive type of skin cancer, with ultraviolet (UV) radiation being a major environmental factor contributing to approximately 86% of melanoma cases [1]. UV-induced DNA damage, primarily in the form of cyclobutene pyrimidines dimer (CPDs), (6−4) pyrimidine-pyrimidone photoproducts (6−4 PPs) bulky lesions

**Data availability statement:** All relevant data are within the manuscript and its Supporting information files.

**Funding:** This study was supported by National Institute of Health R21ES034636 to WC. The funding bodies had no role in study design, data collection, data analysis and interpretation of data and in writing the manuscript.

**Competing interests:** The authors have declared that no competing interests exist.

and strand breaks, triggers cellular responses to repair DNA damage and maintain genomic stability. Melanocytes, the pigment-producing cells in the epidermis, are particularly vulnerable to UV-induced DNA damage. Their ability to properly respond to and repair these DNA lesions is crucial in preventing malignant transformation to cutaneous melanoma.

It has been well established that bulky DNA adducts formed by UV damage are primarily repaired through nucleotide excision repair (NER), which operates via two primary pathways: global genome NER (GG-NER) and transcription-coupled NER (TC-NER) [2]. Other repair mechanisms also play a role in removing UV-induced damage [3]. If these bulky DNA lesions are not corrected, they can block the progression of replication forks, leading to replication stress and contributing to genomic instability. This disruption is a key factor in the onset of various diseases, including cancer [4,5].

The CTC1-STN1-TEN1 (CST) complex, a single-strand DNA (ssDNA) binding protein complex, is known for its involvement in DNA replication and repair. It plays an essential role in stabilizing replication forks and facilitating the recovery of replication stress. Previous studies have established that CST helps prevent genomic instability by antagonizing nascent strand DNA degradation [6,7], promoting base excision repair [8,9], inhibiting repriming at UV damage sites [10], facilitating efficient telomeric DNA replication and preventing excessive telomere lengthening by telomerase [11–17], and promoting non-homologous end joining at double-strand breaks (DSBs) by counteracting end resection [18–20]. We have investigated the role of CST in colorectal tumorigenesis and found that STN1 deficiency promotes colorectal cancer development in young adult mice via inhibiting base excision repair [8]. While its role in maintaining genome stability is well-established, the specific contributions of STN1 to skin tumorigenesis remain poorly understood, and the potential link between STN1 dysfunction and cutaneous melanoma has not been explored.

Recently, we have found that CST localizes at replication forks stalled by UV damage to inhibit PrimPol repriming at UV-stalled forks [10]. Depletion of STN1 or CTC1 leads to increased replication progression and accumulation of ssDNA upon UV exposure in a PrimPol-dependent manner, suggesting that CST deficiency may activate PrimPol-directed repriming at UV damage sites to permit the progression of DNA replication upon encountering UV-induced bulky DNA damage [10]. PrimPol-mediated repriming is an effective replication rescue pathway when forks encounter bulky DNA lesions or when fork reversal is defective [21]. However, PrimPol is an error-prone enzyme [22], and excessive repriming increases mutation incorporation, leading to genome instability. Additionally, PrimPol initiates replication downstream of the DNA lesion by "skipping" it, leaving a ssDNA gap behind [23,24]. Such ssDNA is also a cause of genome instability. Our findings reveal an important role of CST in regulating replication progression at UV damage sites.

Previous GWAS studies reported that STN1 (aka OBFC1) deregulation is associated with melanoma [25,26], implicating a potential role of STN1 or CST in skin tumorigenesis. While *in vitro* studies are vitally important for mechanistic investigation, they cannot recapitulate the complex cellular interactions within the tumor microenvironment or the interactions with the host immune system that play key roles in cancer initiation

and progression. No animal model was currently available to investigate the role of CST in skin tumorigenesis caused by UV exposure. In this study, we have engineered the first tamoxifen-inducible melanocyte-specific knockout (cKO) mouse model, in which STN1 can be depleted specifically in melanocytes in adult mice. After induction of STN1 depletion, mice are exposed to UV irradiation and the effects of STN1 downregulation on skin tumorigenesis are evaluated. Our results show that STN1 downregulation in mature melanocytes has no effect on cutaneous melanoma development. Our findings provide important insights into the molecular mechanisms that regulate skin tumorigenesis and suggest that while CST plays a critical role in maintaining genome stability, its specific contribution to melanoma formation may be more complex than previously anticipated. We discuss the possible interpretations of the results and suggest areas of future research.

## Materials and methods

### CTC1 and STN1 gene expression analysis

Analysis of CST gene alterations was performed with cBioPortal using TCGA PanCancer Atlas Skin Cutaneous Melanoma dataset. CTC1, STN1, TEN1 gene expression in skin cutaneous melanoma was performed with TNMplot [27] using RNA-seq data from the Genotype-Tissue Expression (GTEx) portal and TCGA database.

### Animal handling

All animals were housed and studied in specific pathogen-free animal facilities at Loyola University Chicago Health Sciences campus. All studies were approved by the Institutional Animal Care & Use Committee (IACUC) approval #LU212698. All animal procedures were conducted in strict accordance with institutional and national ethical guidelines, with efforts taken to minimize suffering and distress through the use of appropriate anesthetics, analgesics, and specialized housing conditions as needed. All research staff involved in animal handling completed accredited training programs in animal care and handling prior to the commencement of the study. Euthanasia was performed by $CO_2$ inhalation until 1–2 minutes after cessation of breathing, followed by cervical dislocation. Mice showing injuries or skin ulcerations were treated with a topical antibiotic and analgesic according to veterinarian instructions. Animals showing severe ulceration or distress were euthanized. Anesthesia was done with inhalation of Isoflurane USP (Pivetal 78949580) at 4% induction and 2% maintenance when shaving their fur at the application site to minimize stress to the animal.

### Generation of melanocyte-specific tamoxifen-inducible STN1 cKO mice and genotyping

Wild-type C57BL/6 (stock# 000664) was purchased from the Jackson Laboratory. The *STN1*<sup>*Flox/Flox*</sup> (*STN1*<sup>*F/F*</sup>) C57BL/6 mouse strain was generated and described previously [8]. The C57BL/6 strain expressing *Braf*<sup>*V600E*</sup> under the control of tamoxifen-inducible melanocyte-specific Cre (*Tyr::CreER*<sup>*T2*</sup>; *Braf*<sup>*V600E*</sup>) in which the Cre recombinase is driven by the tyrosinase promoter was described previously [28] and obtained from Dr. Richard Marais. *STN1*<sup>*F/F*</sup> mice were crossed to *Tyr::CreER*<sup>*T2*</sup>; *Braf*<sup>*V600E*</sup> mice to generate *Tyr::CreER*<sup>*T2*</sup>; *STN1*<sup>*F/F*</sup> animals. STN1 deletion in melanocytes was verified by performing genotyping. All animals were weaned at 21–28 days old.

Primers used for genotyping are listed below:

| Primer Name | Sequence |
| --- | --- |
| BRAF 137 | GCTTGGCTGGACGTAAACTC |
| BRAF 125 | GCCCAGGCTCTTTATGAGAA |
| BRAF 143 | AGTCAATCATCCACAGAGACCT |
| CreERT2 LL622 | TGAAGGGTCTGGTAGGATCA |
| CreERT2 LL148 | GAAGCAACTCATCGATTG |
| STN1loxPshift Forward | TGTAATCCCAGCGCTCAGGAG |
| STN1loxPshift Reverse | GATCTGACAGAGATCTCCTGGCT |

 

Genotyping of *STN1^F/F* was performed by PCR using tail snip DNA as described previously [8]. Genotyping of *Tyr::CreER^T2* was performed with primers CreERT2 LL622 and CreERT2 LL148, yielding a PCR product of approximately 450 bp if the gene is present on either allele. For the above PCR reactions, thermocycling conditions consisted of one step of 5 min at 95°C, 32 cycles of 30 sec at 95°C, 30 sec at 55°C, and 1 min at 72°C, followed by 5 min at 72°C. Reactions contained 1 µl of of tail lysate DNA, 0.6 µM primers, 250 µM dNTPs, 2.5 units of KOD Hot Start Polymerase (Sigma/Millipore), 3 mM $MgSO_4$, 5% DMSO, and 10x PCR buffer in a total of 25 µl reaction.

Genotyping of heterozygous *Braf^V600E* was performed with primers BRAF 137, BRAF 125 and BRAF 143, yielding a PCR product of approximately 500 bp for the internal control and approximately 140 bp for the modified allele. For the above PCR reaction, thermocycling conditions consisted of one step of 5 min at 95°C, 32 cycles of 30 sec at 95°C, 30 sec at 55°C, and 1 min at 72°C, followed by 5 min at 72°C. Reactions contained 1 µl of of tail lysate DNA, 0.6 µM of BRAF 137 and BRAF 143 each primer and 0.8 µM BRAF 125 primer, 250 µM dNTPs, 2.5 units of KOD Hot Start Polymerase (Sigma/Millipore), 3 mM $MgSO_4$, 5% DMSO, and 10x PCR buffer in a total of 25 µl reaction.

### UV irradiation and tumor formation

Tamoxifen (Sigma-Aldrich T5648) was freshly prepared in 100% ethanol. At 8 weeks of age, the back skin of animals was shaved, and warm tamoxifen (100 mg/ml) was applied to the shaven skin approximately 24 hours following shaving. Tamoxifen application was repeated at day 3, 5, 8. Twenty-eight days after tamoxifen application, animals were subjected to broad band UVB irradiation (160 mJ/cm$^2$) weekly for up to 10 months using a UVB light source (Analytikjena #95-0042-08) as previously described [29]. All UV exposure were performed at approximately 7:00 AM. The back skin of animals was shaved as needed to expose the skin at least one day prior to UV irradiation. A total of 58 animals were given tamoxifen and exposed to UV. Animals were checked weekly as to their general health status and were more frequently examined when they were subjects of the experiment. The development of skin tumors was monitored over the course of the study. Mice were sacrificed when tumor burden reached ethical limits or if the animals displayed signs of ill health or distress.

### Histological analysis

After the UV exposure protocol, mice were euthanized, and their skin was collected for histopathological analysis. Tissues were fixed in 10% formalin, embedded in paraffin, and sectioned. Skin samples were examined using hematoxylin and eosin (H&E) staining to assess histological features such as keratinocyte morphology, tumor grade, and invasion. All sectioning and H&E staining were performed at Loyola University Stritch School of Medicine Laboratory of Pathology core. Images were captured under an EVOS XL light microscope (Invitrogen) with a 20× objective.

### Immunohistochemical (IHC) staining

IHC staining for Stn1 and S100B was performed to assess the presence of melanoma cells in tissue samples. Briefly, formalin-fixed, paraffin-embedded tissue sections (4 µm thick) were deparaffinized with xylene and rehydrated through a series of graded alcohols. Sections were then incubated with a primary antibody against Stn1 (1:100, Santa Cruz, 376450) and S100B (1:5,000, ProteinTech, 15146-AP) overnight at 4°C. Lastly, samples were counterstained with hematoxylin. Images were captured under an EVOS XL light microscope (Invitrogen) with a 20× objective. For detecting DNA damage and UV lesions, IHC was performed with tissue sections using γ-H2AX (1:500; Active Motif, 39118) and CPD (1:500; Cosmo Bio, NMDND001), respectively. Images were acquired using an EVOS XL light microscope (Invitrogen)

with a 20 × objective. Quantification of γ-H2AX– and CPD–positive cells in tumor regions was performed with ImageJ using the IHC Profiler plugin for nuclear staining analysis.

## Statistical analysis

To calculate the statistical significance of tumor formation between the control and experiment groups for males and females, two-sided Fisher exact tests were used.

## Results

### CTC1, STN1, TEN1 expression is downregulated in human melanoma tissues

Previous GWAS studies have found that STN1 deregulation is associated with melanoma [25,26]. We therefore first analyzed the TCGA database to determine whether CST alteration is associated with skin cancers. Our analysis showed that the combined gene alteration frequency of CTC1, STN1 and TEN1 in human melanoma is ~ 9%, within the range of alteration frequency of some of the well-known melanoma-associated genes like PTEN, p53, and TERT, although NRAS, BRAF are the top altered genes in melanoma [30,31] (Fig 1A). Second, all three CST genes are significantly downregulated in cutaneous melanoma tissues compared to normal [27] (Fig 1B). These observations implicate a potential role of CST deficiency in skin cancer predisposition.

### Generation of melanocyte-specific tamoxifen-inducible *STN1* cKO mice

Our study using cell lines reveals that CST deficiency causes excessive PrimPol repriming upon UV exposure, potentially driving genome instability [10]. To investigate the role of CST in regulating melanomagenesis *in vivo*, an animal model is needed. Since CST is critical for genome replication and stability, it is highly likely that germline knockout of *CST* genes may cause developmental defects or even embryonic lethality. In fact, *CTC1* deleted mice die prematurely due to global cellular proliferative defects caused by complete bone marrow failure [14]. We therefore engineered a *STN1* cKO mouse model. STN1 was chosen because STN1 links the CTC1 and TEN1 subunits by interacting with both of them, but CTC1 and TEN1 do not directly interact [32]. Removing STN1 disrupts the formation of the CST complex and the potential CTC1/STN1 or STN1/TEN1 subcomplexes [17,32], thus is expected to eliminate the potential compensation caused by subcomplex formation.

The *STN1^{F/F}* mice, which we generated in our previous study [8] (Fig 2A), were crossed with *Tyr::CreER^{T2}* transgenic mice, in which Cre recombinase is driven by the tyrosinase promoter to specifically delete STN1 in melanocytes. In

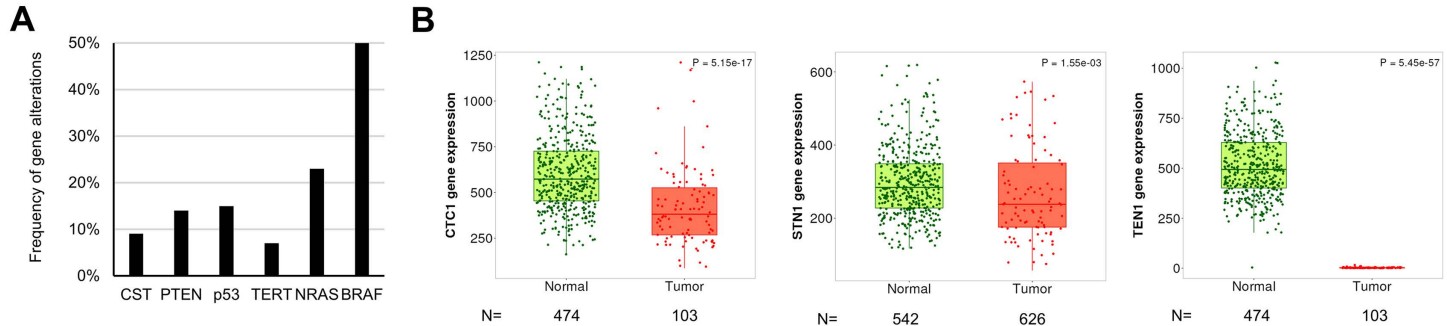

**Fig 1. CST gene alterations and expression in cutaneous melanoma tissues. (A)** Frequency of CST gene alterations observed in cutaneous melanoma. Well-known melanoma-associated genes are shown for comparison. Data are derived from TCGA PanCancer Atlas Skin Cutaneous Melanoma dataset. **(B)** CTC1, STN1, TEN1 expression is significantly downregulated in skin cutaneous melanoma. Analysis was performed using TNMplot based on RNA-seq data from paired tumor and normal samples. *P*: Mann-Whitney U test. *N*: number of samples.

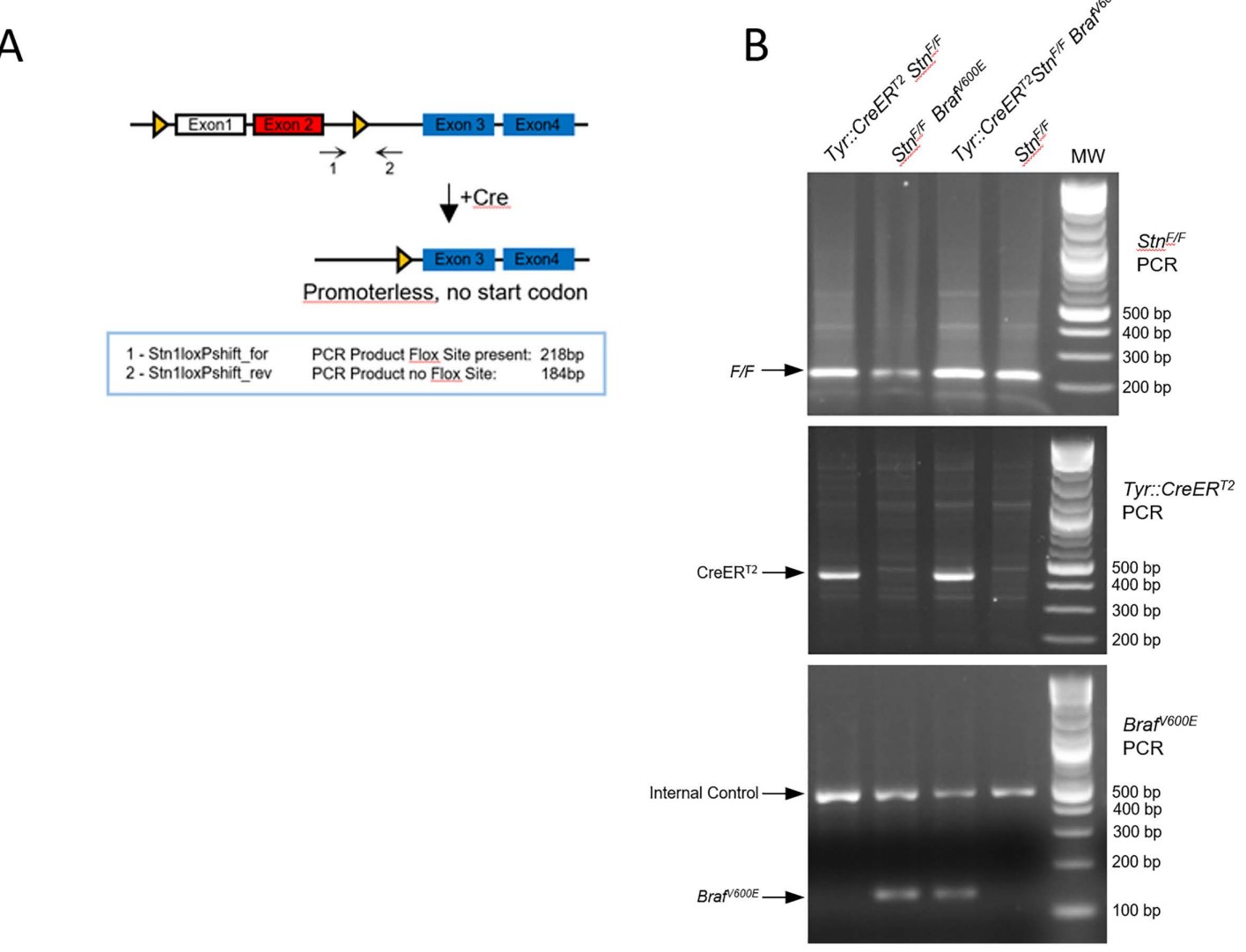

**Fig 2. Generation of melanocyte-specific tamoxifen-inducible *STN1* cKO mice. (A)** Scheme of *STN1* cKO. The murine *STN1* gene contains 10 exons. Only exons 1-4 are shown. The ATG start codon is located in exon 2 (red). Exon 1 is a non-coding exon. Triangles: *loxP* sites. **(B)** Genotyping results of *STN1^{F/F}*, *Tyr::CreER^{T2}* and *Braf^{V600E}* alleles in mice used in this study.

addition, the *CreER^{T2}* allele controls Cre expression through tamoxifen (TAM) administration. Thus, the *Tyr::CreER^{T2}; STN1^{F/F}* mouse model induces STN1 deletion in a time-specific (by TAM administration in adult mice) and tissue-specific (only in melanocytes) manner, allowing STN1 deletion only in mature melanocytes without affecting skin or organismal development. Genotyping confirmed the *STN1^{F/F}* allele and *Tyr::CreER^{T2}* allele (Fig 2B).

## STN1 deficiency has no obvious effect in melanoma incidence in adult mice

TAM was topically applied to shaved skin on the backs of 8-week-old mice to induce *Cre* expression (Fig 3A). IHC staining of skin tissues showed that Stn1 was undetectable in melanocytes in *Tyr::CreER^{T2}; STN1^{F/F}* mice following TAM

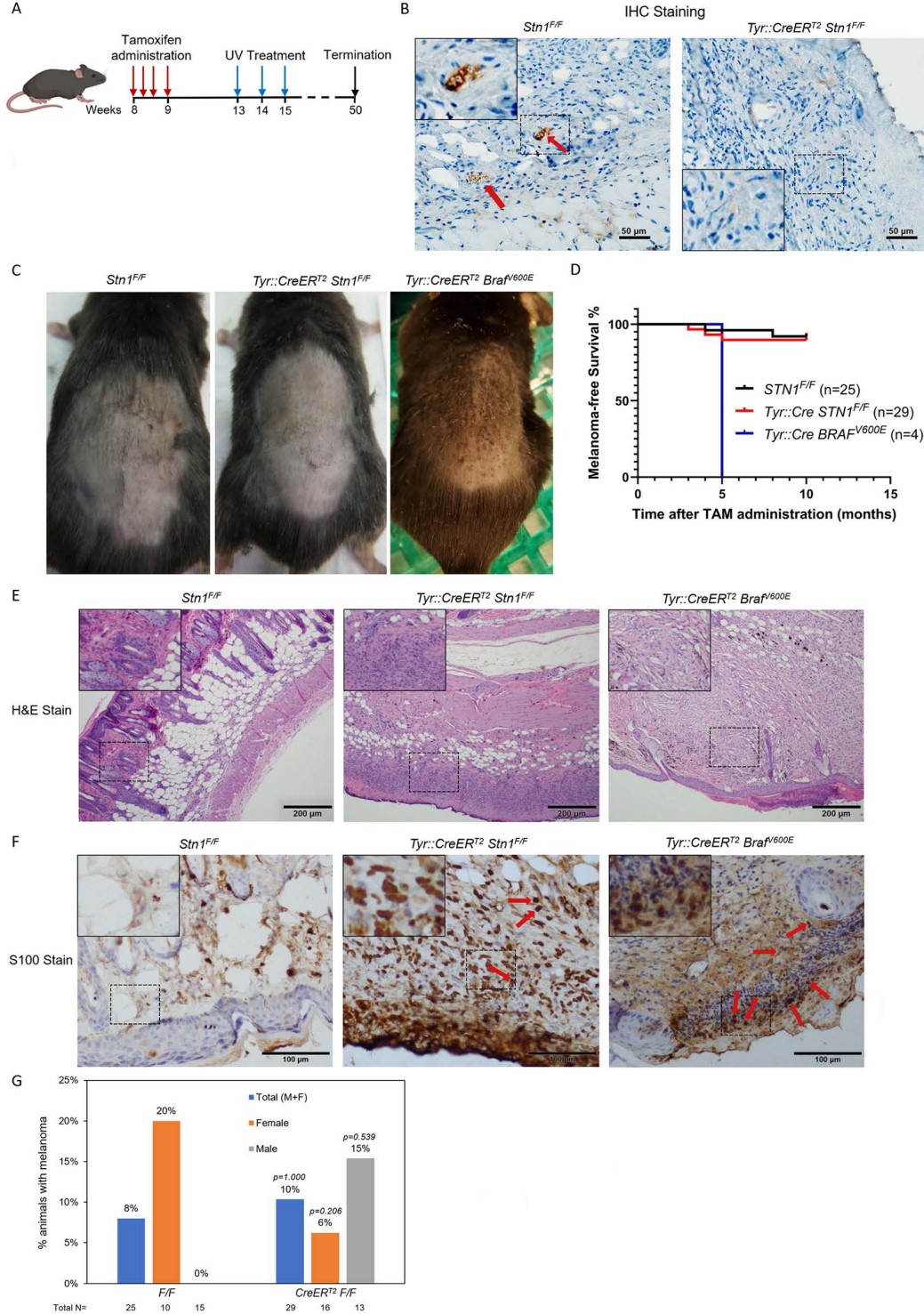

**Fig 3. *STN1* depletion in melanocytes has no obvious effect on melanomagenesis in adult mice. (A)** Scheme of TAM treatment and UV exposure. **(B)** IHC of Stn1 in skin tissue sections from *Tyr::CreER^T2; STN1^F/F* and wild-type animals. Red arrows point to Stn1 staining in hair follicles of the control sample. Inserts show amplified images of the boxed areas. **(C)** Representative image of melanoma (dark speckles) developed on the back of the shaved

skin in *Tyr::CreER*$^{T2}$; *Braf*$^{V600E}$ mice. **(D)** Kaplan-Meier curve showing melanoma-free survival in UV-irradiated animals. **(E)** Representative H&E staining of skin tissues collected from mice. Inserts show amplified images of the boxed areas. **(F)** Representative IHC staining of S100B in skin tissues collected from mice. Inserts show amplified images of the boxed areas. Red arrows point to S100B positively stained cells. **(G)** Melanoma formation frequency in UV-irradiated animals. *P*: two-sided Fisher exact tests. The number of animals in each group is shown below the plot.

administration, while its expression remained robust in the hair follicles of *STN1*$^{F/F}$ controls (Fig 3B), suggesting Stn1 was efficiently depleted by TAM treatment. Stn1 was almost undetectable outside of hair follicles in mouse skin (Fig 3B), likely due to its low expression in the skin compared to other tissues, consistent with its similarly low expression in human skin [33].

Following TAM treatment, mice were irradiated with mild dose of UVA/UVB (160 mJ/cm$^2$) weekly for up to 10 months, and the skin tumor incidence and tumor latency were observed weekly during the treatment course (Fig 3A). Since it has been shown that skin carcinogenicity in mice is regulated by circadian clock [29], all UV exposure were performed at ~7:00 AM to ensure consistent circadian rhythm. As a positive control, we included *Tyr::CreER*$^{T2}$; *Braf*$^{V600E}$ animals. Consistent with previous reports [28], 100% (n = 4) of *Tyr::CreER*$^{T2}$; *Braf*$^{V600E}$ animals developed visible melanoma ~10 weeks after UV exposure, as shown by distinct black pigmentation on the back skin (Fig 3C), with a median latency of 5 months (Fig 3D). In contrast, among the total of 29 STN1-deficient animals (13 males and 16 females), only two male mice and one female animal developed melanoma at 3–5 months of UV exposure. The majority of STN1-deficient mice did not develop visible melanoma even after prolonged UV exposure for 10 months (Fig 3C) and survived prolonged UV exposure (Fig 3D). Control mice (n = 25, 15 males and 10 females) showed similar melanoma incidents, with only two female animals developing melanoma (Fig 3D).

Mice were sacrificed either at the time of stress caused by high tumor load or 10 months after TAM administration. Skin tissues were collected, fixed, sectioned, and stained with H&E (Fig 3E) or IHC. After pathological analysis, skin tissues that were potentially melanoma positive were then stained with the melanoma marker S100B to confirm melanoma formation (Fig 3F). Our result shows that the overall melanoma incidence in STN1-deficient mice is no different from the control littermates (*p* = 1.000, two-sided Fisher exact tests). To ensure that STN1 depletion indeed caused increased genome instability, IHC was performed on collected skin tissues to detect γH2AX and CPD. As shown in Fig 4, STN1-deficient mice displayed increased γ-H2AX and CPD damage, consistent with the role of STN1 in protecting genome stability. Interestingly, the STN1-deficient male cohort showed a moderate increase in melanoma formation and the female STN1-deficient mice showed a reduction in melanoma formation (Fig 3G). However, statistical analysis showed no significance of tumor incidences in male (*p* = 0.539, two-sided Fisher exact tests) or female (*p* = 0.206, two-sided Fisher exact tests) cohort between STN1-deficient mice and control animals (Fig 3G), *p*robably due to the lower number of melanoma-positive animals. Thus, we were unable to make a definitive conclusion about the effect of gender difference in melanoma formation in STN1 deficient animals.

## Discussion

The CST complex plays a crucial role in maintaining genome stability, particularly under replication stress, by stabilizing replication forks and regulating DNA damage repair [6,7,18–20]. We recently found that CST dysfunction could lead to genomic instability through upregulating repriming by PrimPol after UV exposure, which is expected to lead to the accumulation of mutations and ssDNA [10]. In addition, our analysis of the TCGA database revealed that CST genes, including STN1, are frequently downregulated in cutaneous melanoma tissues compared to normal tissues (Fig 1). This observation aligns with previous GWAS studies reporting STN1 deregulation to melanoma predisposition [25,26]. However, its role in skin tumorigenesis, particularly in the context of UV-induced damage, remains poorly understood. In this study, we developed the first melanocyte-specific *STN1* cKO mouse model and investigated the role of STN1 in skin tumorigenesis. Using

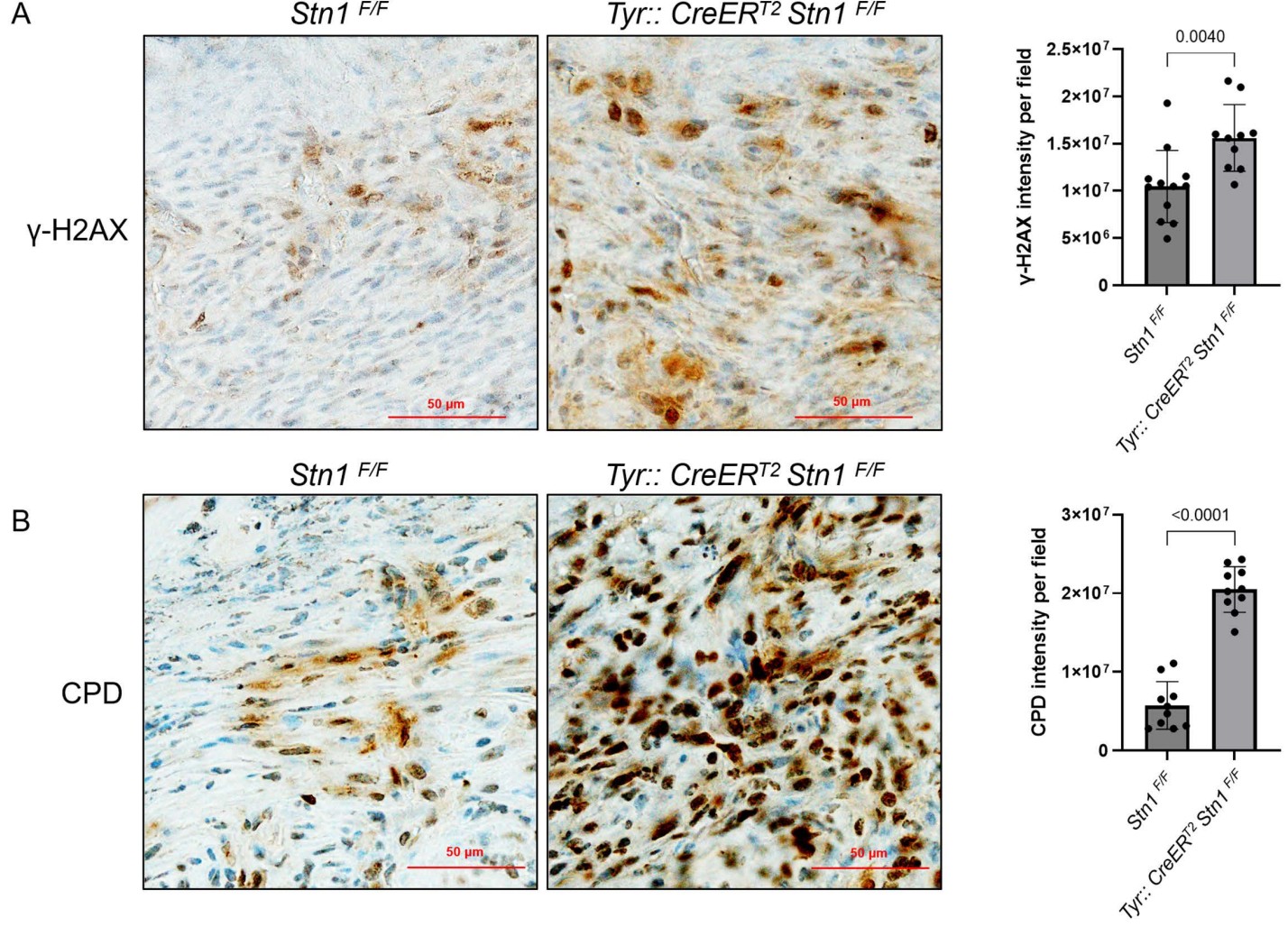

**Fig 4. IHC shows increased CPD and DNA damage levels in melanoma cells upon STN1 depletion. (A)** Representative images of γH2AX IHC staining of melanoma tissues collected from *Stn1F/F* and *Tyr::CreERT2 Stn1F/F* mice. γH2AX signal intensity was quantified using ImageJ and plotted. **(B)** Representative images of CPD IHC staining of melanoma tissues collected from *Stn1F/F* and *Tyr::CreERT2 Stn1F/F* mice. CPD signal intensity was quantified using ImageJ and plotted.

this mouse model, we found that STN1 deficiency had no significant effect on melanoma incidence following chronic UV exposure (Fig 3). These results suggest that while the *in vitro* evidence suggesting that CST deficiency leads to increased replication stress and genomic instability, STN1 loss in mature melanocytes does not appear to significantly alter melanoma susceptibility under the conditions tested in this study.

One possible explanation is that compensatory mechanisms, such as DNA repair pathways or redundancy in the replication stress response, may mitigate the effects of STN1 loss *in vivo*. Additionally, the complex tumor microenvironment and immune surveillance in intact organisms may play a role in suppressing tumorigenesis despite increased genomic instability at the cellular level. Interestingly, our results showed some gender-specific differences in melanoma development, with male STN1-deficient mice showing a slight increase in melanoma formation. Although this difference did not reach statistical significance, it raises the possibility that gender-specific hormonal or immune responses could modulate the effects of

CST dysfunction in melanocytes [1]. The lack of significance may be due to the relatively small sample size and low overall tumor incidence in both groups. Further studies with larger cohorts are needed to determine whether gender-specific factors, such as hormonal influences or differences in DNA repair capacity, contribute to the observed trends.

In contrast to the lack of a significant melanoma phenotype, our previous work demonstrated that STN1 deficiency promotes colorectal cancer (CRC) development in young adult mice [8], suggesting that the tumor-promoting effects of CST dysfunction may be tissue- and context-dependent. In the CRC model, STN1 depletion led to enhanced genomic instability, including increased replication stress and oxidative damage, which promoted tumor formation in the colon. The discrepancy between our findings in melanoma and CRC tumorigenesis may reflect differences in the cellular environment, tissue-specific DNA damage response mechanisms, and tumor microenvironments. While STN1 plays a critical role in maintaining genomic stability in both tissues, the absence of an observable effect in melanoma could be due to compensatory mechanisms that are more prominent in skin cells or the fact that melanocytes are less reliant on STN1 in the context of UV-induced damage. Moreover, the potential for tumorigenesis in CRC may be more sensitive to the replication defects induced by STN1 deficiency, as the colon epithelium is actively proliferating and may be more susceptible to the accumulation of mutations and replication stress. Thus, while our data suggest that STN1 depletion does not significantly alter melanoma formation in this model, the effects of CST deficiency on cancer development may be more pronounced in other tissues with different proliferative rates or exposure to distinct types of DNA damage.

One limitation of our study is the use of a TAM-inducible knockout model, which restricts STN1 deletion to mature melanocytes. While this approach avoids developmental defects associated with germline CST deletion, it may not fully recapitulate the effects of chronic CST deficiency from an early age. Given the critical role of CST in DNA replication and damage repair, CST deficiency during early stages of melanocytes may have different consequences. Future investigation may be needed to understand this. Additionally, the UV exposure protocol used in this study, while sufficient to induce melanoma in the positive control group (*Tyr::CreER*$^{T2}$; *Braf*$^{V600E}$ mice), may not have been intensive enough to reveal subtle effects of STN1 deficiency on skin tumorigenesis. Future studies could explore the effect of STN1 depletion in earlier developmental stages or with more aggressive UV exposure regimens or combine STN1 deficiency with other genetic alterations such as *Tp53, Pten* or *Nras* to better model the complex etiology of skin cancer. Finally, the development of cKO models for other CST components (CTC1 or TEN1) could help elucidate the relative contributions of each subunit to skin tumorigenesis.

Our data show that STN1 depletion in mature melanocytes produces a marginal decrease in melanoma incidence in female mice but a slight increase in males, although the differences are statistically insignificant. This subtle sex difference may reflect hormonal differences between male and female mice that can influence melanoma susceptibility and progression. Estrogen and androgens modulate DNA repair, immune responses, cell proliferation, and gene expression programs related to melanocyte biology [34–36]. It has been shown that 17β-estradiol, the predominant estrogen during a woman's reproductive years, attenuates nucleotide excision repair, the major pathway for repairing DNA damage caused by UV exposure [37]. Estrogen and androgen signaling also regulate the expression and function of key DNA repair proteins, influencing pathways such as homologous recombination and non-homologous end-joining. Estrogen modulates the expression of many key DNA repair genes including BRCA1, BRCA2, MRE11, RAD50, and PALB2 [38–40], which could contribute to more robust genome maintenance and protective effects in females. In contrast, androgen receptor signaling increases the expression and recruitment of DNA-PK and Ku70/Ku80, promoting double-strand break repair, but may also drive tumorigenesis by supporting proliferation and limiting DNA damage sensing in melanoma cells [41–43]. In addition, estrogen signaling has been shown to promote melanocyte differentiation, suppress proliferation, enhance response to immunotherapy, and may confer protective effects against melanoma progression, contributing to improved outcomes in females [44]. In contrast, androgen and androgen receptor signaling can drive melanoma invasiveness, promote immune evasion by suppressing NK cell-mediated cytotoxicity, and contribute to resistance against targeted therapies [45,46]. These sex hormone–mediated differences in DNA repair capacity, along with variations in immune surveillance, may partly explain the modest and divergent effects observed in males and females following Stn1 loss.

Our findings suggest that STN1 may not be a critical driver of UV-induced melanoma in adult mice. However, this does not rule out the possibility that CST dysfunction contributes to other types of skin cancer or plays a role in tumor progression rather than initiation. For example, CST deficiency may influence the response to chemotherapy or radiation therapy, which also induce replication stress and DNA damage. Further research is needed to explore these possibilities and to determine whether CST components could serve as therapeutic targets in specific contexts.

This study provides the first *in vivo* evidence that STN1 deficiency in melanocytes does not significantly impact UV-induced melanoma development in adult mice, despite *in silico* associations between CST dysfunction and melanoma. While CST dysfunction has been implicated in genomic instability and replication stress, our findings suggest that its role in skin tumorigenesis may be context-dependent or compensated by other mechanisms *in vivo*. These results highlight the complexity of DNA damage response pathways and underscore the need for further research to fully understand the contributions of CST to cancer etiology.

## Supporting information

**S1 Fig. Uncropped raw images used in Figs 2 and 4.** (A) Uncropped gel images for Fig 2B. (B) Original uncropped IHC images. Boxed areas indicate those used for quantification in Fig 4.
(PDF)

**S1 Table. The values used to build graphs in Fig 4.**
(XLSX)

## Acknowledgments

We are grateful for Dr. Richard Marais at Cancer Research UK Manchester Institute for sharing the *Tyr::CreER^T2^; Braf^V600E^* animal. We thank Dr. Yu-Ying He at University of Chicago and Dr. Mitchell Denning at Loyola University Chicago SSOM for helpful discussion.

## Author contributions

**Conceptualization:** Weihang Chai.

**Data curation:** Sara Knowles, Fang Wang, Weihang Chai.

**Formal analysis:** Sara Knowles, Fang Wang, Maarten C. Bosland, Weihang Chai.

**Funding acquisition:** Weihang Chai.

**Investigation:** Sara Knowles, Fang Wang.

**Methodology:** Sara Knowles, Fang Wang.

**Project administration:** Weihang Chai.

**Resources:** Weihang Chai.

**Supervision:** Weihang Chai.

**Visualization:** Sara Knowles, Fang Wang.

**Writing – original draft:** Sara Knowles, Fang Wang, Weihang Chai.

**Writing – review & editing:** Maarten C. Bosland, Shobhan Gaddameedhi, Weihang Chai.

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
