## [Decision Letter · Decision Letter 0]

14 Aug 2025

In vivo investigation of STN1 downregulation in melanoma formation in adult mice following UV irradiation

PLOS ONE

Dear Dr. Chai,

Thank you for submitting your manuscript to PLOS ONE. After careful consideration, we feel that it has merit but does not fully meet PLOS ONE’s publication criteria as it currently stands. Therefore, we invite you to submit a revised version of the manuscript that addresses the points raised during the review process.

We look forward to receiving your revised manuscript.

Kind regards,

Rishi Jaiswal, Ph.D.

Academic Editor

PLOS ONE

Journal Requirements:

2. To comply with PLOS One submissions requirements, in your Methods section, please provide additional information regarding the experiments involving animals and ensure you have included details on (1) methods of sacrifice, (2) methods of anesthesia and/or analgesia, and (3) efforts to alleviate suffering.

4. We note that your Data Availability Statement is currently as follows: All relevant data are within the manuscript and in Supporting Information files.

Additional Editor Comments:

This manuscript investigates whether STN1, a component of the CST complex important for genome stability, plays a role in UV-induced melanoma formation in vivo. The authors first show that CST genes are downregulated and sometimes altered in human melanoma datasets. They then generate a melanocyte-specific, tamoxifen-inducible STN1 conditional knockout mouse and chronically expose these mice to UV radiation. Contrary to expectations from in vitro studies, STN1 loss in mature melanocytes does not significantly alter melanoma incidence compared to controls. The work provides the first in vivo test of CST function in UV-driven melanoma but finds no clear effect under the experimental conditions. I have two major concerns.

1- IHC shows reduced Stn1 staining, but no functional assays (e.g., markers of replication stress, DNA damage) are provided to confirm that melanocytes lacking STN1 in vivo are indeed experiencing the expected genomic instability under UV stress. Such assays would strengthen the link between the negative tumor result and the molecular role of STN1.

2- The knockout is induced in mature melanocytes. This leaves open the possibility that STN1 loss during melanocyte development, or in earlier life stages, might have different consequences. The restriction to adult deletion could mask effects relevant to melanoma initiation. Authors should acknowledge this limitation more explicitly.

Reviewers' comments:

Reviewer's Responses to Questions

**Comments to the Author**

1. Is the manuscript technically sound, and do the data support the conclusions?

Reviewer #1: Yes

Reviewer #2: Yes

2. Has the statistical analysis been performed appropriately and rigorously?

Reviewer #1: No

Reviewer #2: Yes

3. Have the authors made all data underlying the findings in their manuscript fully available?

Reviewer #1: Yes

Reviewer #2: No

4. Is the manuscript presented in an intelligible fashion and written in standard English?

Reviewer #1: Yes

Reviewer #2: Yes

Reviewer #1: The paper requires more thorough statistical analysis on graphs (some is there). Additionally, the figures I was able to view were very low resolution making it more difficult to assess. I think they'v made an interesting observation, and I am interested in the gender differences they observe. I think it is worth speculating in one in two sentences based on literature potential examples of differential dna repair or influe4nces of hormones on replication stress and repair.

Reviewer #2: Comments

The authors have generated a Tamoxifen-inducible melanocyte specific Stn1 knock-out mouse model to study melanoma formation after UV exposure. They have used Braf-mutant animals as control for their experiments. The authors have previously shown that Stn1 deficiency makes mice susceptible to colorectal cancers, and the current study precisely shows that it might not be the case with melanoma. Although Stn1 is known to be downregulated in human melanoma patients, this might not be true in case of mouse models. This is a robust study and helps explain concerns in Stn1 mediated melanoma. I find this study suitable for publication after a few of my concerns are answered-

Major

All the previous in-vitro experiments have been performed in human cell lines. Based on these results the authors performed experiments in mouse models. It would be better if they could study the role of Stn1 in cell proliferation and DNA repair in primary melanocytes isolated from these animals under UV-exposure. The extent of DNA damage induced in melanocytes should be measured by gamma-H2AX or CPD levels. This may explain the observed phenotype.

**Do you want your identity to be public for this peer review?** For information about this choice, including consent withdrawal, please see our Privacy Policy

Reviewer #1: No

Reviewer #2: No

---

## [Author Response · Author response to Decision Letter 1]

22 Oct 2025

We have addressed editor's and reviewers' concerns. Our detailed response is all included in the Point-to-point response file. Thank you for reviewing our manuscript.

---

## [Editor Report · Decision Letter 1]

28 Oct 2025

In vivo investigation of STN1 downregulation in melanoma formation in adult mice following UV irradiation

PONE-D-25-30140R1

Dear Dr. Chai,

We’re pleased to inform you that your manuscript has been judged scientifically suitable for publication and will be formally accepted for publication once it meets all outstanding technical requirements.

Kind regards,

Academic Editor

PLOS ONE

Additional Editor Comments (optional):

Dear Dr. Chai,

Thank you for submitting the revised version of your manuscript entitled “In vivo investigation of STN1 downregulation in melanoma formation in adult mice following UV irradiation.”

I have carefully reviewed your responses to the reviewers’ and editor’s comments, as well as the revised manuscript. I am pleased to note that you have thoroughly addressed all the concerns raised in the previous round of review. The revisions have significantly improved the clarity and overall quality of the manuscript.

Based on the current version, I am satisfied that the manuscript meets the publication standards of PLOS ONE. I am therefore pleased to recommend the paper for acceptance.

Congratulations on your careful and comprehensive revision, and thank you for choosing PLOS ONE for the publication of your work.

Best regards,

Academic Editor

PLOS ONE
---

## [Editor Report · Acceptance letter]

PONE-D-25-30140R1

PLOS ONE

Dear Dr. Chai,

I'm pleased to inform you that your manuscript has been deemed suitable for publication in PLOS ONE. Congratulations! Your manuscript is now being handed over to our production team.

Kind regards,

on behalf of

Dr. Rishi Jaiswal

Academic Editor

PLOS ONE